# Beyond Visual cues: Harnessing Text signal for Test-Time OOD Detection

## Abstract

Out-of-Distribution (OOD) detection has become increasingly critical for deploying reliable machine learning systems in open-world environments. While vision-language models (VLMs) like CLIP demonstrate strong potential for OOD detection, most existing test-time OOD detection methods focus on storing representative visual features, leaving the textual modality's adaptation potential largely unexplored. In this work, we investigate whether text-side adaptation can improve test-time OOD detection. To this end, we propose Test-time Textual OOD Discovery (TTOD), a framework that harnesses semantic knowledge directly from the test data stream with an unknown distribution. Our method progressively constructs a retrievable OOD textual knowledge bank by continuously updating OOD prompts during testing under the guidance of pseudo labels from a base detector. To alleviate the impact of contaminated signals, the method further develops a purification strategy that exploits clustering properties of similar OOD types to separate ID samples misclassified as OOD by the base detector from OOD samples, thereby improving pseudo-label quality for more effective adaptation. Extensive experiments on two standard benchmarks with nine OOD datasets demonstrate that TTOD consistently achieves state-of-the-art performance, highlighting the value of textual intervention for robust test-time OOD detection.

## 1 Introduction

Deep learning demonstrates remarkable performance in closed-set scenarios where training and testing data follow identical distributions. However, when deployed in open-world environments, models frequently encounter OOD data from unknown classes. Critically, models often misclassify such OOD samples as high-confidence in-distribution (ID) classes (Nguyen et al., 2015; Hendrycks & Gimpel, 2017), posing significant safety risks in critical applications such as autonomous driving and medical diagnosis. Therefore, accurately detecting OOD data is essential for ensuring the reliability and safety of AI systems in real-world deployments.

Traditional OOD detection methods (Hendrycks & Gimpel, 2017; Hendrycks et al., 2022; Liu et al., 2020; Sun et al., 2021) rely on well-trained ID classifiers but are limited to the visual modality. The emergence of pre-trained VLMs like CLIP (Radford et al., 2021) has enabled leveraging multi-modal information for enhanced OOD detection. Recent VLM-based methods have been dedicated to learn OOD-related knowledge through external images or text labels (Wang et al., 2023; Jiang et al., 2024), or extract such knowledge solely from ID training data, such as background regions (Miyai et al., 2023) or randomly cropped images (Zeng et al., 2025). However, OOD features derived from specific datasets inherently cannot cover the infinitely diverse OOD data found in the real world.

In order to obtain more practical OOD knowledge, recent methods employ test-time adaptation to adapt VLMs to true OOD distributions. One straightforward method (Cao et al., 2025) trains the OOD detector using test data labeled by a base OOD detector (Figure 1a). However, this method relies solely on updating model parameters and fails to retain historically discriminative decision boundaries—ultimately leaving the detector without global discriminative capabilities. To preserve useful historical information, while some methods only store visual features observed during testing (Yang et al., 2025), AdaNeg (Zhang & Zhang, 2024) additionally leverages the textual modality by aligning external OOD text semantics with actual test distributions, achieving stronger performance. However, AdaNeg relies on a finite set of OOD texts to represent the infinite OOD spectrum,

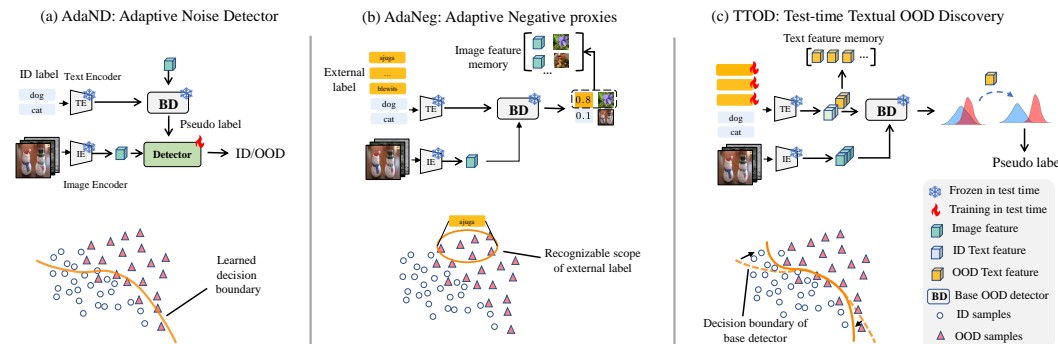

Figure 1: Comparison of different test-time OOD detection methods. For a single test batch, (a) AdaND achieves online adaptation by training an OOD detector; (b) AdaNeg stores representative image features with external labels; (c) Our method leverages autonomous text knowledge discovery to enhance historical information utilization without external labels. It dynamically discovers and improves OOD prompts before storing the refined representations in a self-updating prompt bank.

which proves inherently inadequate. This limitation results in poor adaptability to OOD samples beyond the scope of predefined categories (Figure 1b), leading to suboptimal performance on challenging benchmarks. Inspired by prompt learning (Zhou et al., 2022), fine-tuning prompts enables text embeddings to better align with actual data distributions. We naturally pose a question:

*Would directly learning OOD textual semantics from the test stream—rather than aligning specific ones with OOD distributions—yield better adaptation results?*

To this end, this study proposes Test-time Textual OOD Discovery (TTOD), a novel framework that harnesses textual knowledge directly from test streams without relying on external labeled datasets. As illustrated in Figure 1c, the framework builds upon a synergistic pipeline with three key components. First, *OOD Knowledge Discovering* gets a pseudo-label of the test sample by using a base detector and optimizes learnable prompts to extract OOD textual representation. Seconds, *OOD Knowledge Purification* employs a novel strategy: leveraging the clustering properties of similar OOD types to separate ID boundary samples from the potential OOD set, improving pseudo-label quality for more effective prompt supervision. These discoveries are then stored in an *OOD Textual Knowledge Bank* that maintains a dynamic repository of high-quality embeddings for robust score calibration across evolving test batches, enabling continuous adaptation without external supervision. Our contributions can be summarized as follows:

- We propose TTOD, the first test-time textual OOD discovery framework that eliminates reliance on external labeled datasets by dynamically extracting discriminative OOD textual knowledge directly from test streams, enabling robust adaptation to diverse and previously unseen OOD scenarios.

- We propose a novel purification strategy that exploits the clustering properties of OOD types in the semantic space to separate ID boundary samples from the potential OOD set, effectively filtering contaminated signals from pseudo-label.

- Comprehensive experiments across multiple benchmarks show that TTOD consistently achieves state-of-the-art performance, with significant improvements of 11.63% FPR95 and 4.29% AUROC over existing methods on average.

## 2 RELATED WORK

**OOD Detection.** Traditional OOD detection methods focus on single-modal image analysis, falling into two categories. The first designs scoring functions using model outputs (e.g., logits, features, layer statistics) (Hendrycks & Gimpel, 2017; Liang et al., 2018; Liu et al., 2020; Sun et al., 2021). The second explores ID-OOD decision boundaries via various training strategies (Du et al., 2022; Ming et al., 2023; Chen et al., 2024b;a). Though achieving satisfactory results, they overlook textual modalities' rich semantic information, leading to sub-optimal performance (Jiang et al., 2024).

To leverage textual knowledge, recent research has focused on employing vision-language models like CLIP (Radford et al., 2021) with powerful multimodal understanding capabilities. These VLM-based approaches can be categorized into three main strategies. Concept matching methods such as MCM (Ming et al., 2022) and CMA (Lee et al., 2025) leverage CLIP's image-text alignment to generate OOD scores based on category names or auxiliary concepts. GL-MCM (Miyai et al., 2025) extends MCM to multi-object scenarios. ID-enhanced methods improve discrimination by exploiting additional information from ID data. For example, FA (Lu et al., 2025) uses ID prompts as references for learnable prompt optimization; LoCoOp (Miyai et al., 2023) and SCT (Yu et al., 2024) employ entropy maximization to reduce background sensitivity; OSPCoOp (Xu et al., 2025), IDLike (Bai et al., 2024), Negprompt (Li et al., 2024), and Local-Prompt (Zeng et al., 2025) generate pseudo-OOD samples through background extraction, image cropping, or assuming the relationship between OOD distribution and ID distribution. External knowledge-based methods tackle OOD detection using explicit OOD information. For example, Neglabel (Jiang et al., 2024) collects potential OOD labels from large-scale corpora, while CLIPN (Wang et al., 2023) learns negative prompts from massive datasets. However, such methods prove impractical due to the inherent diversity and infinite nature of real-world outliers.

In order to obtain more practical OOD knowledge, there has been growing interest in leveraging information from real-time testing scenarios to assist OOD detection. AUTO (Yang et al., 2023) updates the parameters of all batch normalization layers and the final feature block in the model by reducing prediction confidence for potential OOD samples. Unlike updating the original model, AdaND (Cao et al., 2025) freezes the classification model and detects OOD samples by training an additional noise detector. OODD (Yang et al., 2025) maintains a priority queue to accumulate more representative OOD image features, which are used to calibrate detector outputs for test samples. AdaNeg (Zhang & Zhang, 2024) further exploits external textual labels to guide the selection and storage of visual features, achieving stronger OOD detection performance. Despite these advances, existing test-time methods focus primarily on visual-side adaptation or on using text only as a fixed auxiliary cue. In contrast, our method actively discovers discriminative OOD textual knowledge during testing, directly exploiting the adaptive potential of the text modality.

**Prompt Learning.** Originated in NLP as a replacement for manual prompt engineering, prompt learning has been adapted to vision-language scenarios with VLMs like CLIP (Radford et al., 2021) serving as strong baselines. Methods such as CoOp (Zhou et al., 2022) use learnable vectors for template words, boosting CLIP's performance across tasks. While applied to OOD detection (Bai et al., 2024; Lu et al., 2025; Zeng et al., 2025), existing approaches operate in training-time with labeled ID data. Ours is the first to apply it to test-time OOD detection, adaptively learning textual knowledge aligned with real deployment OOD distributions—enabling dynamic adaptation without pre-defined OOD categories or external datasets.

## 3 METHODOLOGY

### 3.1 PRELIMINARIES

**OOD detection.** In OOD detection, the goal is to accurately categorize ID samples into their respective classes and reject OOD samples as non-ID. For any test sample $\mathbf{x}$, this goal can be formulated as a binary classification task using a scoring mechanism $S(\cdot)$ to differentiate between ID and OOD inputs with the decision threshold $\lambda$. The decision function $D(\mathbf{x})$ is defined as:

$$D(\mathbf{x}) = \begin{cases} ID, & \text{if } S(\mathbf{x}) \geq \lambda \\ OOD, & \text{if } S(\mathbf{x}) < \lambda \end{cases}. \tag{1}$$

**CLIP and threshold determination.** For CLIP-based OOD detection, it leverages both visual and textual modalities for improved discrimination. CLIP uses two pre-trained encoders: an image encoder $f(\cdot)$ that converts input image $\mathbf{x}$ into a feature vector $\mathbf{z} = f(\mathbf{x}) \in \mathbb{R}^d$, and a text encoder $g(\cdot)$ that converts class prompt $\mathbf{u}_c$ into a feature vector $\mathbf{t}_c = g(\mathbf{u}_c) \in \mathbb{R}^d$, where $\mathbf{u}_c$ represents the text prompt for class $c$. The cosine similarity $\cos(\mathbf{z}, \mathbf{t}_c)$ measures the degree of image-text alignment. For CLIP-based OOD detection, following Li et al. (2023a), the threshold $\lambda$ is adaptively determined by minimizing intra-class variance based on the bimodal distribution of OOD scores:

$$\min_{\lambda} \frac{1}{N_{\text{id}}} \sum_{S(\mathbf{x}_i) > \lambda} [S(\mathbf{x}_i) - \mu_{\text{id}}]^2 + \frac{1}{N_{\text{ood}}} \sum_{S(\mathbf{x}_j) \leq \lambda} [S(\mathbf{x}_j) - \mu_{\text{ood}}]^2, \tag{2}$$

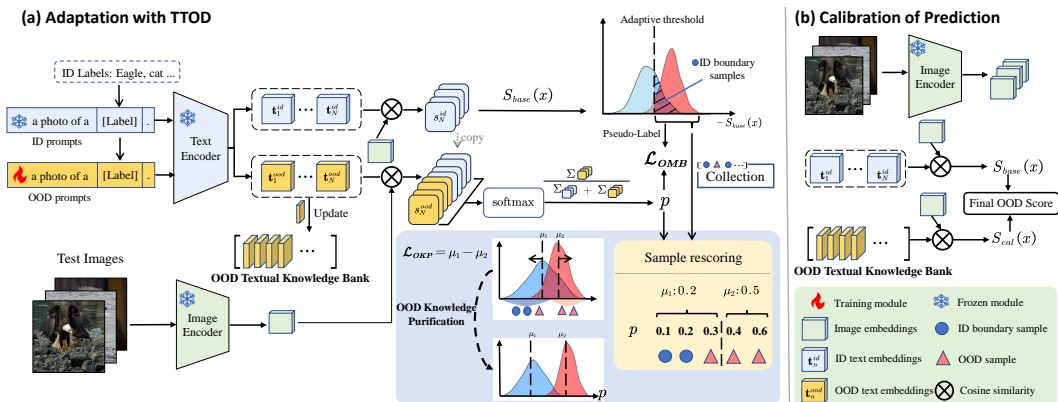

Figure 2: Overview of the proposed TTOD. Guided by pseudo labels obtained from a base detector $S_{base}(\mathbf{x})$, OOD knowledge discovery is performed. To filter out contaminated signals within pseudo labels, an OOD Knowledge Purification strategy is employed. Finally, the learned OOD textual embeddings are updated into the OOD Textual Knowledge Bank for real-time prediction calibration.

where $\mu_{\text{id}} = \frac{1}{N_{\text{id}}} \sum_{S(\mathbf{x}_i) > \lambda} S(\mathbf{x}_i)$ and $\mu_{\text{ood}} = \frac{1}{N_{\text{ood}}} \sum_{S(\mathbf{x}_j) \leq \lambda} S(\mathbf{x}_j)$ are the mean scores above and below threshold $\lambda$, respectively. $N_{\text{id}}$ and $N_{\text{ood}}$ denote the number of samples above and below the threshold in a test-time queue of length $N_q$.

## 3.2 TEST-TIME TEXTUAL OOD DISCOVERY

Existing test-time adaptive OOD detection methods for VLMs operate exclusively in the visual modality by caching or processing image features. Given that VLMs learn cross-modal representations through text-image alignment, we propose to discover OOD textual knowledge related to the actual distribution during test time, rather than relying on stored visual features. As illustrated in Figure 2, our method consists of three key components that exploit textual representations. First, *OOD Knowledge Discovery* employs $\mathcal{L}_{OMB}$ (OOD-Focused Minority Balanced Loss) to optimize learnable text prompts by balancing sample weights between ID and OOD classes during pseudo-label based training. Second, *OOD Knowledge Purification* employs $\mathcal{L}_{OKP}$ to filter misclassified ID samples by exploiting the clustering property of similar OOD types in the textual embedding space. Finally, *OOD Textual Knowledge Bank* maintains a dynamic repository of refined textual embeddings for robust score calibration across evolving test batches. By operating in the textual space rather than on visual features, our pipeline achieves continuous adaptation to distribution shifts without external supervision. The overall optimization objective for test-time adaptation incorporates both discovery and purification: $\mathcal{L} = \mathcal{L}_{\text{OMB}} + \alpha \cdot \mathcal{L}_{\text{OKP}}$, where $\alpha$ controls the trade-off between learning from pseudo-labeled data and purifying knowledge quality.

**OOD Knowledge Discovering.** The objective is to discover discriminative textual embeddings that effectively separate ID and OOD samples during test time, without relying on ground-truth ID/OOD labels. To achieve this, a pseudo-labeling strategy is adopted where a base detector provides weak supervision signals to guide the optimization of OOD prompts toward semantically meaningful representations. Specifically, the OOD prompts are first initialized with the ID prompts template. Then, for each batch of incoming samples, text embeddings $\mathbf{t}_k^{\text{ood}}$ and $\mathbf{t}_j^{\text{id}}$ are obtained from the OOD prompts and ID prompts, respectively, along with image embeddings $\mathbf{z}$ for the test image. A base OOD detector $S_{base}(\cdot)$ is used to obtain the OOD score $S_{base}(\mathbf{x})$ of the sample $\mathbf{x}$, after which an adaptive threshold $\lambda$ is applied to obtain the pseudo label $\hat{y}$:

$$\hat{y} = \begin{cases} 0 & \text{if } S_{\text{base}}(\mathbf{x}) < \lambda \\ 1 & \text{if } S_{\text{base}}(\mathbf{x}) \geq \lambda \end{cases}, \tag{3}$$

where $\hat{y} = 1$ indicates ID samples, and $\hat{y} = 0$ indicates OOD samples. To integrate the learnable OOD prompts into the detection mechanism, the OOD prediction probability of $\mathbf{x}_i$ is estimated by

measuring the alignment between these textual embeddings and the image embedding, i.e.,

$$p(\mathbf{x}_i) = \sum_{k=1}^{M} q_k(\mathbf{x}_i) = \frac{\sum_{k=1}^{M} \exp\left(\cos(f(\mathbf{x}_i), \mathbf{t}_k^{\text{ood}})/\tau\right)}{\sum_{j=1}^{N} \exp\left(\cos(f(\mathbf{x}_i), \mathbf{t}_j^{\text{id}})/\tau\right) + \sum_{j=1}^{M} \exp\left(\cos(f(\mathbf{x}_i), \mathbf{t}_j^{\text{ood}})/\tau\right)}, \quad (4)$$

where $M$ denotes the number of learnable OOD textual prompts, $q_k(\mathbf{x}_i)$ represents the response intensity of the $k$-th OOD embedding to the $i$-th sample $\mathbf{x}_i$. Summing the response intensities of all OOD prompts to integrate diverse OOD semantics avoids biases from any single prompt. The probability $p(\mathbf{x}_i)$ and pseudo-labels $\hat{y}$ together provide supervision for optimizing the learnable prompts. To address the challenge of severe class imbalance commonly encountered in test-time scenarios, an OOD-focused minority balanced loss is designed to ensure balanced learning from both minority and majority classes:

$$\mathcal{L}_{\text{OMB}} = -\frac{1}{\pi_+} \mathbb{E}_{(\mathbf{x}, \hat{y}) \sim \mathcal{D}_{\text{test}}} \left[\mathbf{1}\{\hat{y} = 1\} \log(1 - p(\mathbf{x}))\right] - \frac{1}{\pi_-} \mathbb{E}_{(\mathbf{x}, \hat{y}) \sim \mathcal{D}_{\text{test}}} \left[\mathbf{1}\{\hat{y} = 0\} \log p(\mathbf{x})\right], \quad (5)$$

where $\pi_+$ and $\pi_-$ denote the proportions of samples labeled as potential ID and OOD, respectively, estimated using empirical frequencies within mini-batches.

**OOD Knowledge Purification.** While the learned textual embeddings provide valuable OOD textual knowledge, their quality is diminished by contamination signals from pseudo-labeling. Since perfect OOD detectors rarely exist (AUROC < 1), samples flagged as OOD inevitably include a mixture of OOD samples and misclassified ID samples—we term these misclassified samples as "ID boundary samples". Such a mixture not only limits the discriminative power of the learned embeddings but also risks disrupting subsequent adaptation processes.

To address this challenge, we leverage a key property of CLIP's image-text alignment training: text prompts serve as compact, concept-level representations with higher-level semantic structure, which gives rise to a critical "semantic clustering" phenomenon. Specifically, a single text prompt corresponding to a certain semantic concept can map to multiple OOD image samples sharing that concept, forming a cluster in the feature space. This semantic clustering property enables us to filter pseudo-label noise by exploiting the shared textual semantics inherently present in test streams. The mechanism works as follows. When most OOD samples in a batch belong to the same high-level textual semantic cluster (denoted as semantic A), if the base detector misclassifies some samples in this cluster as ID—while the textual semantics of the actual ID samples in the batch (denoted as semantic B) are distinctly different from semantic A—we can leverage this semantic discrepancy to rectify the misclassifications. Specifically, samples labeled OOD are first collected, and these samples are then re-scored using the OOD probability score $p(\mathbf{x})$ computed via the OOD prompts, which quantifies their alignment with semantic A. Meanwhile, the OOD prompts are further optimized to strengthen their affinity with the semantic A cluster while pushed away from the confusing semantic B cluster. This process effectively filters out contaminated supervision signals caused by misclassified samples. Building on this text-driven mechanism, OOD knowledge purification is achieved by explicitly separating high-confidence OOD samples from uncertain ones within each batch and encouraging their OOD probability distributions to become more bimodal. Formally, the purification process can be obtained by minimizing the OOD knowledge purification loss

$$\mathcal{L}_{\text{OKP}} = -\left(\frac{1}{|S_h|} \sum_{i \in S_h} p(\mathbf{x}_i) - \frac{1}{|S_\ell|} \sum_{j \in S_\ell} p(\mathbf{x}_j)\right), \quad (6)$$

where $S_h = \{i \mid p(\mathbf{x}_i) > \theta\}$ represents purified OOD samples with higher confidence, while $S_\ell = \{j \mid p(\mathbf{x}_j) \leq \theta\}$ denotes the uncertain set containing ID boundary samples and ambiguous OOD instances. Here, $\theta$ is the adaptive threshold computed for OOD pseudo-labeled samples using OOD probability $-p(\mathbf{x})$ as OOD scores. The gradient formula for $\mathcal{L}_{\text{OKP}}$ with respect to OOD embedding $t_k^{ood}$ is as follows:

$$\nabla_{t_k^{ood}} \mathcal{L}_{\text{OKP}} = -\left(\frac{1}{|S_h|} \sum_{i \in S_h} q_k(\mathbf{x}_i)(1 - p(\mathbf{x}_i))\mathbf{z}_i - \frac{1}{|S_\ell|} \sum_{i \in S_\ell} q_k(\mathbf{x}_i)(1 - p(\mathbf{x}_i))\mathbf{z}_i\right), \quad (7)$$

where $q_k(\mathbf{x}_i)$ denotes the response intensity of the k-th OOD embedding to sample $i$. This update drives $t_k^{ood}$ closer to high-confidence OOD samples while further away from uncertain samples. The

per-sample weight $q_k(\mathbf{x}_i)(1 - p(\mathbf{x}_i))$ ensures updates focus on samples relevant to $k$-th prompt and down-weights saturated examples, making the optimization both class-specific and adaptive.

**OOD Textual Knowledge Bank.** A key challenge in test-time adaptation is that text embeddings optimized on individual batches capture only local semantic patterns, making them vulnerable to distribution shifts in subsequent batches. To overcome this limitation, we propose the OOD Textual Knowledge Bank (OKB), which accumulates discriminative textual embeddings discovered across test batches. The OKB serves two critical purposes: (1) preserving valuable OOD textual knowledge that might be forgotten when adapting to new batches, and (2) providing broader semantic coverage through diverse distribution patterns. By leveraging cross-batch semantic consistency through such historical knowledge, more robust OOD detection is enabled. To maintain computational efficiency, the bank operates with a fixed capacity $K$. Each OOD textual embedding is evaluated using a potential OOD score, defined as its minimum distance to all ID textual embeddings:

$$S_{\text{in}}(\mathbf{t}_i^{ood}) = \min_c \left[ -\cos(\mathbf{t}_c^{id}, \mathbf{t}_i^{ood}) \right]. \tag{8}$$

This score guides the bank's update strategy: when capacity is reached, the method retains only the $K$ highest-scoring embeddings, ensuring the most discriminative OOD patterns are preserved.

For OOD prompt initialization, we focus adaptation on challenging near-OOD samples by initializing prompts in the immediate neighborhood of ID embeddings via the same textual template (e.g., "a photo of a {classname}"). For testing computational efficiency, a shared prompt prefix (instead of class-specific ones) is used to enable scalable adaptation across diverse test scenarios.

### 3.3 CALIBRATION OF THE PREDICTION

The OOD Knowledge Bank calibrates the base detector using real OOD semantic information from testing. For each test sample with feature $\mathbf{z}$, the final score combines base detection with semantic calibration:

$$S_{\text{final}}(\mathbf{x}) = S_{\text{base}}(\mathbf{x}) + \beta \cdot S_{\text{cal}}(\mathbf{x}), \quad \text{where} \quad S_{\text{cal}}(\mathbf{x}) = - \max_{j \in \{1, \ldots, K\}} \cos(\mathbf{z}, \mathbf{t}_j^{ood}) \tag{9}$$

The negative sign in $S_{\text{cal}}(\mathbf{x})$ ensures samples similar to stored OOD semantics receive lower scores, while $\beta$ balances semantic information and order of magnitude. This mechanism continuously refines OOD detection by leveraging accumulated real OOD semantic information.

## 4 EXPERIMENTS

### 4.1 EXPERIMENTAL SETUP

**Datasets and evaluation protocol.** Following prior protocols (Miyai et al., 2023; Chen et al., 2024b), we evaluate our method on two standard benchmarks. For large-scale evaluation, ImageNet-1K (Deng et al., 2009) was used as the ID dataset, with OOD test sets comprising iNaturalist (Horn et al., 2018), SUN (Xiao et al., 2010), Places (Zhou et al., 2018), and Texture (Cimpoi et al., 2014). Other experiment was also conducted using CIFAR-100 (Krizhevsky et al., 2009) as the ID dataset to enable broader comparison across different scales and resolutions, with corresponding OOD datasets including SVHN (Netzer et al., 2011), LSUN-C (Yu et al., 2015), LSUN-R (Yu et al., 2015), iSUN (Xu et al., 2015), Texture (Cimpoi et al., 2014), and Places365 (Zhou et al., 2018). For evaluation, we adopt two standard metrics: (1) FPR95: false positive rate at 95% recall; (2) AUROC: area under the receiver operating characteristic curve

**Implementation details.** Following previous studies (Zhang & Zhang, 2024), we use CLIP (Radford et al., 2021) with ViT-B/16 (Dosovitskiy et al., 2021) as our VLM and MCM (Ming et al., 2022) as the base OOD detector. The OOD prompt is optimized using the AdamW (Kingma & Ba, 2015) optimizer with a learning rate of 0.005 and batch size of 64. TTOD has four hyperparameters: loss weight $\alpha = 0.5$ for balancing training objectives, OKB capacity $K = 2048$ for storing text feature, threshold queue length $N_q^M = 4096$ for adaptive threshold $\theta$, and ID dataset-specific fusion coefficient $\beta$ for integrating MCM scores (0.006 for CIFAR-100, 0.0005 for ImageNet-1k, set based on score distributions). All experiments run on a single Nvidia 3090 GPU, with additional details provided in the Appendix.

Table 1: Performance comparison on ImageNet-1k OOD detection benchmarks. The results marked with $^\dagger$ are taken from (Lu et al., 2025). Lower FPR95 and higher AUROC are better. Best results are in **bold** and the second-best results are underlined.

| Method | iNaturalist | | SUN | | Places | | Texture | | **Average** | |
|---|---|---|---|---|---|---|---|---|---|---|
| | FPR95↓ | AUROC↑ | FPR95↓ | AUROC↑ | FPR95↓ | AUROC↑ | FPR95↓ | AUROC↑ | FPR95↓ | AUROC↑ |
| *Training-free & non-adaptive methods* | | | | | | | | | | |
| MCM (Ming et al., 2022) | 30.92 | 94.61 | 37.59 | 92.57 | 44.71 | 89.77 | 57.85 | 86.11 | 42.77 | 90.76 |
| GL-MCM (Miyai et al., 2025) | 15.09 | 96.72 | 29.08 | 93.41 | 37.07 | 90.37 | 58.94 | 83.11 | 35.04 | 90.90 |
| CMA (Lee et al., 2025) | 23.84 | 96.89 | 30.11 | 93.69 | 29.86 | 93.17 | 47.35 | 88.47 | 32.79 | 93.05 |
| Neglabel (Jiang et al., 2024) | 2.00 | 99.47 | 20.95 | 95.47 | 36.48 | 91.56 | 45.00 | 90.02 | 26.1 | 94.13 |
| *Training-based methods* | | | | | | | | | | |
| MSP$^\dagger$ (Hendrycks & Gimpel, 2017) | 74.57 | 77.74 | 76.95 | 73.97 | 79.72 | 72.18 | 73.66 | 74.84 | 74.98 | 76.22 |
| ODIN$^\dagger$ (Liang et al., 2018) | 98.93 | 57.73 | 88.72 | 78.42 | 87.80 | 76.88 | 85.47 | 71.49 | 90.23 | 71.13 |
| Energy$^\dagger$ (Liu et al., 2020) | 64.98 | 87.18 | 46.42 | 91.17 | 57.40 | 87.33 | 50.39 | 88.22 | 54.80 | 88.48 |
| ReAct$^\dagger$ (Sun et al., 2021) | 65.57 | 86.87 | 46.17 | 91.04 | 56.85 | 87.42 | 49.88 | 88.13 | 54.62 | 88.37 |
| CLIPN (Wang et al., 2023) | 19.17 | 96.17 | 26.43 | 94.02 | 32.26 | 92.62 | 41.23 | 90.12 | 30.21 | 93.19 |
| LoCoOp (Miyai et al., 2023) | 23.24 | 95.27 | 31.56 | 93.76 | 38.55 | 91.19 | 43.43 | 90.28 | 34.19 | 92.62 |
| IDLike (Bai et al., 2024) | 19.23 | 96.70 | 54.15 | 87.64 | 56.63 | 85.86 | 34.69 | 91.90 | 41.17 | 90.52 |
| NegPrompt$^\dagger$ (Li et al., 2024) | 37.79 | 90.49 | 32.11 | 92.25 | 35.52 | 91.16 | 43.93 | 88.38 | 37.34 | 90.57 |
| Local-Prompt (Zeng et al., 2025) | 8.62 | 98.06 | 23.78 | 95.22 | 32.43 | 92.50 | 48.47 | 88.84 | 28.32 | 93.65 |
| FA (Lu et al., 2025) | 13.37 | 96.8 | 28.83 | 93.12 | 30.3 | 92.54 | 30.50 | 92.66 | 25.75 | 93.78 |
| *Test-time adaptation methods* | | | | | | | | | | |
| AdaNeg (Zhang & Zhang, 2024) | 0.9 | 99.69 | 11.57 | 96.97 | 35.16 | 93.69 | 29.27 | 94.34 | 19.22 | 96.17 |
| OODD (Yang et al., 2025) | 2.13 | 99.39 | 21.99 | 95.01 | 41.91 | 88.13 | 28.53 | 93.84 | 23.64 | 94.09 |
| AdaND (Cao et al., 2025) | 3.45 | 99.1 | 17.07 | 95.86 | 20.95 | 94.57 | **21.88** | 93.00 | 15.99 | 95.63 |
| TTOD (Ours) | **0.42** | **99.87** | **7.18** | **98.45** | **15.86** | **96.22** | 26.39 | **94.6** | **12.46** | **97.29** |

**Comparison methods.** Comparisons of TTOD with competitive CLIP-based OOD detection methods across three paradigms: (1) training-free & non-adaptive methods (MCM, GL-MCM, Neglabel and CMA), (2) training-based methods (MSP, ODIN, Energy, ReAct, CLIPN, LoCoOp, IDLike, NegPrompt, Local-Prompt and FA), and (3) test-time adaptation methods (OODD, AdaNeg and AdaND).

## 4.2 Performance on Benchmarks

**Effectiveness on ImageNet-1k.** As shown in Table 1, TTOD substantially outperforms the best methods in each category: 13.64% lower FPR95 than Neglabel (best non-adaptive method), 13.29% lower than FA (best training-based method), and 3.53% lower than AdaND (best test-time adaptation method). Notably, unlike AdaNeg/Neglabel, TTOD uses no external datasets, yet achieves better performance by discovering task-specific OOD knowledge during testing. This reduces ID-OOD distribution overlap (Figure 3) as TTOD leveraging the extracted OOD textual knowledge for calibration. These results support that TTOD's ability to discover OOD textual knowledge during testing leads to better alignment with the actual data distribution.

**Effectiveness on CIFAR-100.** To further validate our method's generalization, TTOD is evaluated on the CIFAR-100 benchmark. As shown in Table 2, TTOD consistently outperforms all competing methods across individual OOD datasets and achieves the best average performance. Specifically, among test-time adaptation methods, TTOD demonstrates strong performance on the challenging Places365 dataset, where other adaptation approaches struggle to achieve effective detection.

Table 2: Performance comparison on the CIFAR-100 OOD benchmarks.

| Method | SVHN | | LSUN-R | | LSUN-C | | iSUN | | Texture | | Places365 | | **Average** | |
|---|---|---|---|---|---|---|---|---|---|---|---|---|---|---|
| | F↓ | A↑ | F↓ | A↑ | F↓ | A↑ | F↓ | A↑ | F↓ | A↑ | F↓ | A↑ | F↓ | A↑ |
| Neglabel | 54.33 | 85.13 | 35.63 | 87.43 | 28.22 | 92.47 | 35.72 | 87.69 | 52.97 | 89.45 | 91.46 | 63.12 | 41.82 | 81.01 |
| FA | 15.58 | 97.33 | 48.02 | 89.75 | 33.11 | 93.18 | 51.29 | 89.37 | 22.34 | 95.47 | 46.28 | 89.49 | 36.11 | 92.43 |
| AdaNeg | 16.8 | 95.13 | 32.29 | 89.85 | 30.84 | 91.35 | 38.09 | 88.1 | 45.88 | 91.31 | 79.21 | 78.08 | 40.52 | 88.97 |
| OODD | 60.6 | 90.52 | 75.2 | 84.61 | 44.72 | 92.08 | 75.98 | 83.85 | 90.85 | 73.12 | 98.09 | 59.09 | 74.24 | 80.55 |
| AdaND | 3.51 | 99.12 | 11.29 | 96.87 | 6.11 | 98.24 | 17.36 | 95.6 | 19.96 | 93.53 | 74.37 | 70.65 | 22.1 | 92.34 |
| TTOD (Ours) | **0.01** | **99.98** | **2.52** | **99.47** | **2.01** | **99.54** | **1.27** | **99.67** | **1.68** | **98.91** | **6.65** | **97.98** | **2.36** | **99.26** |

### 4.3 ANALYSIS OF OUR PROPOSED METHOD

**Ablation study.** Ablation studies are conducted on ImageNet-1k to validate the key components of TTOD: $\mathcal{L}_{OMB}$, $\mathcal{L}_{OKP}$, and OKB. As shown in Table 3, $\mathcal{L}_{OMB}$ provides the foundation by discovering OOD-related textual knowledge. The differential loss $\mathcal{L}_{OKP}$ further enhances performance by better separating ID-boundary and OOD samples. The OKB component demonstrates comparable importance when combined with $\mathcal{L}_{OMB}$, indicating that accumulating historical embeddings captures more discriminative OOD knowledge. The full TTOD framework, combining all components, achieves optimal performance, demonstrating the effectiveness of its synergistic integration.

Table 3: Effectiveness of each component.

| $\mathcal{L}_{OMB}$ | $\mathcal{L}_{OKP}$ | OKB | FPR95↓ | AUROC↑ |
|---|---|---|---|---|
| ✗ | ✗ | ✗ | 42.77 | 90.76 |
| ✓ | ✗ | ✗ | 30.56 | 92.54 |
| ✓ | ✓ | ✗ | 24.59 | 93.95 |
| ✓ | ✗ | ✓ | 18.40 | 95.63 |
| ✓ | ✓ | ✓ | **12.46** | **97.29** |

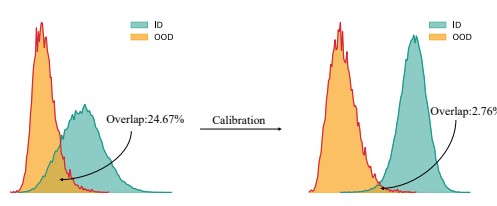

Figure 3: Comparison of the MCM score distribution on SUN dataset.

**Calibration strategies.** Four strategies for enerating final calibrated prediction $S_{final}$ using discovered OOD knowledge are evaluated on ImageNet-1k: (1) Max OOD Similarity (MaxSim); (2) Exponentiated Sum OOD (ExpSum); (3) ID-OOD Softmax Ratio (IDR); and (4) Base-Calibration Score (BCS). As shown in Table 4, the most effective strategy is BCS, which directly subtracts OKB similarity from the base detector score. This show that OOD knowledge functions as a complementary corrective signal, with effectiveness depending primarily on knowledge quality rather than sophisticated scoring mechanisms. Detailed formulations are provided in the Appendix.

**Variants of $\mathcal{L}_{OMB}$.** $\mathcal{L}_{OMB}$ consistently outperforms standard cross-entropy $\mathcal{L}_{CE}$ (Table 5) on ImageNet-1k. By balancing the weight of ID/OOD sample weights, $\mathcal{L}_{OMB}$ effectively preserving the learning signal of minority-class samples.

**Visual analysis of $\mathcal{L}_{OKP}$.** As shown in Figure 5, with $\mathcal{L}_{OKP}$, the potential OOD set progressively splits into a much clearer bimodal distribution over time. This confirms that $\mathcal{L}_{OKP}$ successfully separates OOD samples from ID boundary samples, the source of contamination, ultimately enabling more reliable test-time adaptation for OOD detection.

Table 4: Comparision of calibration strategies.

| Function | FPR95↓ | AUROC↑ |
|---|---|---|
| MaxSim | 20.52 | 95.57 |
| ExpSum | 19.61 | 95.44 |
| IDR | 22.85 | 94.74 |
| BCS | **12.46** | **97.29** |

Table 5: Ablation study of the $L_{OMB}$.

| Function | FPR95↓ | AUROC↑ |
|---|---|---|
| $\mathcal{L}_{CE}$ | 14.23 | 96.98 |
| $\mathcal{L}_{OMB}$ | **12.46** | **97.29** |

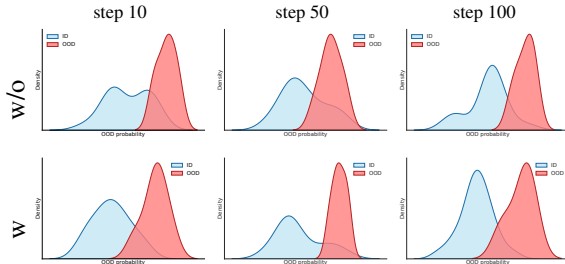

Figure 4: OOD probability density of test samples with OOD pseudo-labels (without/with $\mathcal{L}_{OKP}$, with 64 mixed test samples per step)

**Retention Strategies of OKB.** Four OKB renewal strategies are evaluated on ImageNet-1k: (1) Random (RAND) (Genkin et al., 2023) — random replacement when OKB is full; (2) First-In-First-Out (FIFO) (Yang et al., 2021); (3) Storing All (SA) (Cai et al., 2022) — unlimited retention; and (4) Distance-Based Retention (DBR) (Yang et al., 2025) — keeping entries farthest from ID embeddings. Results in Table 6 show that DBR performs best. Intuitively, DBR keeps the most semantically distinct entries from ID prompts, yielding stronger and more compact corrective signals.

**Visual analysis of OKB.** Figure 5 presents a t-SNE visualization in the cross-modal feature space. While the ID text embeddings lie close to both ID images and some OOD image clusters (i.e.,

they do not clearly separate the two groups), the learned OOD text embeddings in the OKB move noticeably closer to OOD image clusters. Such a shift supports that TTOD successfully discovries textual signals that align with true OOD semantics.

Table 6: Performance of OKB retention strategies.

| Strategy | ImageNet-1k | | CIFAR-100 | |
|---|---|---|---|---|
| | FPR95↓ | AUROC↑ | FPR95↓ | AUROC↑ |
| RAND | 27.29 | 93.07 | 29.06 | 87.07 |
| FIFO | 14.69 | 96.40 | 8.15 | 98.78 |
| SA | 23.19 | 94.27 | 27.33 | 88.04 |
| DBR | **12.46** | **97.29** | **2.36** | **99.26** |

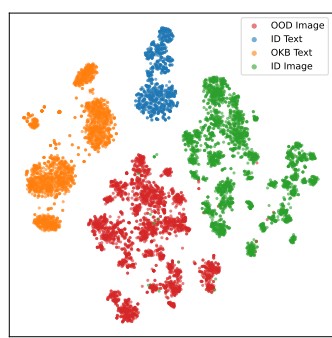

Table 7: OOD prompt learning strategies.

| Prefix | Classname | Update | FPR95↓ | AUROC↑ |
|---|---|---|---|---|
| Random | Random | ✓ | 13.63 | 96.75 |
| Random | ID | ✓ | 15.32 | 96.45 |
| Manual | ID | ✗ | 56.49 | 87.32 |
| Manual | ID | ✓ | **12.46** | **97.29** |

Figure 5: t-SNE visualization of textual embeddings in OKB, utilizing ImageNet-1k, SUN as ID, and OOD dataset respectively.

**OOD prompt learning.** Table 7 compares several OOD-prompt initialization and update schemes on ImageNet-1k. Random represents using random initialization. ID represents using ID classnames initialization. The manual represents using "a photo of a" initialization. The results indicate that initializing OOD prompts with the same textual template as the ID prompts (e.g. "a photo of a classname.") and enabling online optimization at test time yields the most discriminative OOD text embeddings and the best detection performance.

**OOD detector-agnostic.** As Figure 6 shows, TTOD improves performance across all base OOD detectors on ImageNet-1k, demonstrating its detector-agnostic nature. The gains are particularly pronounced when combined with FA, achieving 5.88% FPR95 and 98.76% AUROC, highlighting how test-time textual discovery can effectively complement strong other OOD detectors.

**Sensitivity study.** Sensitivity studies are conducted to evaluate the influence of key hyperparameters: knowledge bank length $K \in [2^8, 2^{14}]$, threshold queue length $N_q^M \in [2^8, 2^{14}]$, loss weighting parameter $\alpha \in [0.2, 1.0]$, and Fusion-Gap Coefficient $\beta \in [0.0002, 0.0008]$. As shown in Figure 7, our method consistently outperforms the strongest baseline across all configurations.

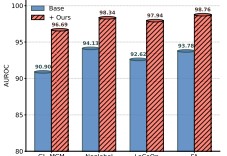

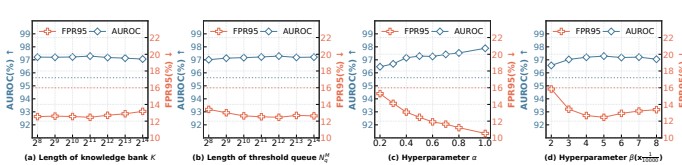

Figure 6: AUROC of TTOD integrated with other base detectors.

Figure 7: Sensitivity studies on ImageNet-1k.

## 5 CONCLUSION

We identify that existing cross-modal OOD detection methods fail to fully exploit the adaptive potential of textual modalities during test time. To address this limitation, we propose TTOD, which progressively constructs a retrievable OOD textual knowledge bank by continuously updating OOD prompts under the guidance of pseudo labels from a base detector To mitigate contamination from imperfect pseudo-labeling, our method incorporates a purification strategy that exploits clustering properties of similar OOD types to separate ID boundary samples, improving pseudo-label quality and adaptation. Notably, TTOD use no external labeled data yet achieves SOTA performance across multiple benchmarks. Extensive experiments validate the effectiveness of each proposed component and demonstrate the value of textual intervention for robust test-time OOD detection.

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

# A  BASIC STATEMENT

## A.1  THE USE OF LARGE LANGUAGE MODELS

Throughout the entire work, we use GPT, Tongyi Qianwen, and Doubao for language polishing and code assistance.

## A.2  REPRODUCIBILITY STATEMENT

Comma separated list of kTo ensure the reproducibility of the experimental results presented in this work, we will make the complete implementation code, including model training configurations and evaluation scripts, publicly available on GitHub upon the acceptance of this paper. Additionally, we will provide detailed documentation specifying dependencies and step-by-step instructions to replicate all reported experiments. This ensures that researchers can easily verify our findings and build upon our method for future work.

## A.3  DETAILED EXPERIMENT SETTING

**Experimental details** The proposed TTOD method was implemented using Python 3.9 and PyTorch 2.3.0, with all experiments conducted on a single NVIDIA GeForce RTX 3090 GPU. Following prior work (Zeng et al., 2025; Shu et al., 2022; Li et al., 2023b; Ming et al., 2022; Yang et al., 2025), we adopted ViT-B/16 (Dosovitskiy et al., 2021) as the backbone model. The OOD prompt was optimized via AdamW (Kingma & Ba, 2015) with a learning rate of 0.005 and batch size of 64.

Table 8: Inference strategies.

| Function | formula | FPR95↓ | AUROC↑ |
|---|---|---|---|
| MaxSim | $-\max_{j\in\{1,...,K\}}\cos(\mathbf{z}_x, \mathbf{t}_j^{ood})$ | 20.52 | 95.57 |
| ExpSum | $-\sum_{j=1}^{K}\exp\left(\cos(\mathbf{z}, \mathbf{t}_i^{\text{ood}})/\tau\right)$ | 19.61 | 95.44 |
| IDR | $\frac{\sum_{j=1}^{N}\exp\left(\cos(\mathbf{z},\mathbf{t}_i^{\text{id}})/\tau\right)}{\sum_{j=1}^{N}\exp\left(\cos(\mathbf{z},\mathbf{t}_j^{\text{id}})/\tau\right)+\sum_{j=1}^{K}\exp\left(\cos(\mathbf{z},\mathbf{t}_j^{\text{ood}})/\tau\right)}$ | 22.85 | 94.74 |
| BCS | $S_{\text{base}}(\mathbf{x}) - \beta \cdot S_{\text{cal}}(\mathbf{x})$ | **12.46** | **97.29** |

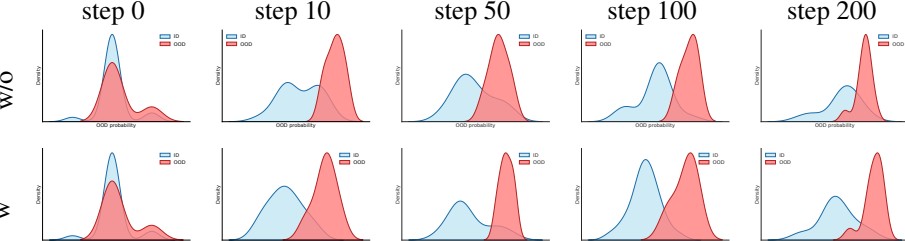

Figure 8: Effect of $\mathcal{L}_{\text{OKP}}$ on separating ID-boundary and OOD samples, 64 samples/step. Top row: OOD probability density without $\mathcal{L}_{\text{OKP}}$; bottom row: OOD probability density with $\mathcal{L}_{\text{OKP}}$.

We set $N_q = 512$ and $\tau = 1$, and used MCM (Ming et al., 2022) as the base OOD detector across all experiments.

TTTOD includes four hyperparameters: (1) loss weight $\alpha = 0.5$; (2) OKB capacity $K = 2048$; (3) threshold queue length $N_q^M = 4096$ (for computing threshold $\theta$); and (4) an ID dataset-specific fusion coefficient $\beta$. Specifically, $\beta$ is set to 0.006 for CIFAR-100 and 0.0005 for ImageNet-1k, determined based on the MCM score distribution.

**Specific inference formula.** Please refer to Table 8.

**More details about Adaptive Threshold.** OWTTT (Li et al., 2023a) searches for the optimal parameter $\lambda$ using a fixed step size of 0.01 between 0 and 1, which is actually unsuitable for different OOD scores (e.g., using ImageNet as ID data and the SUN dataset as OOD data, a step size of 0.01 could span the entire range of MCM Scores across all samples). Here, we propose using the minimum score encountered during testing as the lower bound for the search and the maximum score as the upper bound. Within this range, we uniformly divide the interval into segments matching OWTTT's approach to search for the optimal parameter $\lambda$.

# B ALGORITHMS

This section provides a detailed breakdown of Algorithm 1 (Test-time Textual OOD Discovery), complementing the core description in the main text. The algorithm aims to dynamically adapt OOD text prompts during testing, leveraging both visual and textual information to enhance OOD detection performance without relying on external pre-defined OOD categories.

# C FULL RESULTS OF ABLATION STUDIES

**Effect of $\mathcal{L}_{\text{OKP}}$.** More visualization results about Effect of $mathcalL_{\text{OKP}}$ on separating ID-boundary and OOD samples can be found in Figure 8.

**Different base OOD detectors.** Please refer to Table 9. We can see that we have achieved a positive improvement in performance for different basic detectors. And it's for all datasets, not just the average results.

**Effectiveness of each component.** Please refer to Table 10. We can see that each module is highly effective.

---

**Algorithm 1** Algorithm for Test-time Textual OOD Discover

---

**Require:** test data stream $\{x_i\}_{i=1}^T$, text encoder $g(\cdot)$, image encoder $f(\cdot)$, text prompt for ID $u_c^{id}$,
   learnable text prompt for OOD $u_c^{ood}$, batch size $B$, the priority queue $OKB$ with capacity $K$
1: Initialize OOD prompt: $u_c^{ood} = u_c^{id}$
2: Compute and obtain ID text embeddings: $t_c^{id} = g(u_c^{id})$
3: **for** each data sample $x_i \in \mathcal{D}$ **do**
4:     Calculate base OOD score for $x_i$ using the base detector: $s_{base}(x_i)$
5:     Compute adaptive threshold: $\lambda$ by Equal 2
6:     Assign pseudo-label to $x_i$: $\hat{y}$
7:     Obtain current OOD text embeddings: $t_c^{ood} = g(u_c^{ood})$
8:     Compute OOD probability $p$ by Equal 4
9:     Update queue $Q$: $\mathcal{Q} \leftarrow \mathcal{Q} \cup \{\hat{y}, p\}$
10:    **if** $len(Q) = B$ **then**
11:       train $u_c^{ood}$:
12:       Calculate loss $\mathcal{L}_{OMB}$ loss by Equal 5, input data: $Q$
13:       Compute grouping threshold $\theta$ by Equal 2
14:       Calculate loss $\mathcal{L}_{OKP}$ by Equal 6, input data: $Q, \theta$
15:       Update $u_c^{ood}$ using $\mathcal{L}_{OMB}$ and $\mathcal{L}_{OKP}$
16:       Obtain updated OOD prompt embeddings: $t_c^{ood} = g(u_c^{ood})$
17:       Score the updated OOD prompt embeddings by Equal 8
18:       Store each OOD prompt embedding into the priority queue $OKB$
19:       $Q \leftarrow \emptyset$
20:    **end if**
21:    Perform inference correction and prediction:
22:    Compute final OOD score $S_{final}(x_i)$ by Equal 9
23: **end for**
24: **return** $S_{final}(x_i)$ for all samples in $\mathcal{D}$

---

Table 9: Complementarity to other OOD detectors with the ID dataset of ImageNet-1k. Green indicates an improvement, while red indicates the opposite.

| Method | iNaturalist | | SUN | | Places | | Texture | | **Average** | |
|---|---|---|---|---|---|---|---|---|---|---|
| | FPR95↓ | AUROC↑ | FPR95↓ | AUROC↑ | FPR95↓ | AUROC↑ | FPR95↓ | AUROC↑ | FPR95↓ | AUROC↑ |
| GL-MCM | 15.09 | 96.72 | 29.08 | 93.41 | 37.07 | 90.37 | 58.94 | 83.11 | 35.04 | 90.90 |
| **+ TTOD** | 0.42 | 99.88 | 7.71 | 98.37 | 16.14 | 96.04 | 33.67 | 92.1 | 14.49 | 96.6 |
| Improve | -14.67 | +3.16 | -21.37 | +5.67 | -20.93 | +5.67 | -25.27 | +8.99 | -20.55 | +5.7 |
| Neglabel | 2.00 | 99.47 | 20.95 | 95.47 | 36.48 | 91.56 | 45.00 | 90.02 | 26.1 | 94.13 |
| **+ TTOD** | 0.44 | 99.86 | 6.49 | 98.68 | 15.26 | 96.81 | 10.74 | 98.0 | 8.23 | 98.34 |
| Improve | -1.56 | +0.39 | -14.46 | +3.21 | -21.22 | +5.25 | -34.26 | +7.98 | -17.87 | +4.21 |
| LoCoOp | 23.24 | 95.27 | 31.56 | 93.76 | 38.55 | 91.19 | 43.43 | 90.28 | 34.19 | 92.62 |
| **+ TTOD** | 0.22 | 99.93 | 4.92 | 98.88 | 13.91 | 96.51 | 16.12 | 96.44 | 8.79 | 97.94 |
| Improve | -23.02 | +4.66 | -26.64 | +5.32 | -24.64 | +5.32 | -27.31 | +6.16 | -25.4 | +5.32 |
| FA | 13.37 | 96.80 | 28.83 | 93.12 | 30.30 | 92.54 | 30.50 | 92.66 | 25.75 | 93.78 |
| **+ TTOD** | 0.28 | 99.90 | 5.87 | 98.68 | 10.20 | 97.81 | 7.15 | 98.66 | 5.88 | 98.76 |
| Improve | -13.09 | +3.10 | -22.96 | +5.56 | -20.10 | +5.27 | -23.35 | +6.00 | -19.87 | +4.98 |

Table 10: Effectiveness of each component.

| $\mathcal{L}_{OMB}$ | $\mathcal{L}_{OKP}$ | OKB | iNaturalist | | SUN | | Places | | Texture | | Average | |
|---|---|---|---|---|---|---|---|---|---|---|---|
| | | | FPR95↓ | AUROC↑ | FPR95↓ | AUROC↑ | FPR95↓ | AUROC↑ | FPR95↓ | AUROC↑ | FPR95↓ | AUROC↑ |
| ✗ | ✗ | ✗ | 30.92 | 94.61 | 37.59 | 92.57 | 44.71 | 89.77 | 57.85 | 86.11 | 42.77 | 90.76 |
| ✓ | ✗ | ✗ | 2.62 | 99.39 | 30.81 | 94.88 | 34.95 | 90.68 | 53.85 | 85.22 | 30.56 | 92.54 |
| ✓ | ✓ | ✗ | 1.13 | 99.64 | 16.71 | 97.04 | 25.04 | 94.47 | 55.46 | 84.65 | 24.59 | 93.95 |
| ✓ | ✗ | ✓ | 1.34 | 99.61 | 13.58 | 97.50 | 22.12 | 93.91 | 36.56 | 91.50 | 18.40 | 95.63 |
| ✓ | ✓ | ✓ | 0.43 | 99.87 | 6.48 | 98.55 | 15.29 | 96.38 | 28.19 | 94.02 | **12.46** | **97.29** |

Table 11: Inference strategies.

| Function | iNaturalist | | SUN | | Places | | Texture | | Average | |
|---|---|---|---|---|---|---|---|---|---|---|
| | FPR95↓ | AUROC↑ | FPR95↓ | AUROC↑ | FPR95↓ | AUROC↑ | FPR95↓ | AUROC↑ | FPR95↓ | AUROC↑ |
| MaxSim | 1.04 | 99.66 | 16.23 | 97.24 | 23.07 | 95.21 | 41.74 | 90.17 | 20.52 | 95.57 |
| ExpSum | 0.92 | 99.63 | 14.99 | 97.20 | 22.31 | 95.16 | 40.21 | 89.74 | 19.61 | 95.44 |
| IDR | 1.06 | 99.57 | 17.75 | 96.89 | 23.58 | 95.03 | 49.02 | 87.45 | 22.85 | 94.74 |
| BCS | 0.43 | 99.87 | 6.48 | 98.55 | 15.29 | 96.38 | 28.19 | 94.02 | **12.46** | **97.29** |

Table 12: Comparing $\mathcal{L}_{\mathrm{CE}}$ and $\mathcal{L}_{\mathrm{OMB}}$ across multiple datasets.

| Function | iNaturalist | | SUN | | Places | | Texture | | Average | |
|---|---|---|---|---|---|---|---|---|---|---|
| | FPR95↓ | AUROC↑ | FPR95↓ | AUROC↑ | FPR95↓ | AUROC↑ | FPR95↓ | AUROC↑ | FPR95↓ | AUROC↑ |
| $\mathcal{L}_{\mathrm{CE}}$ | 2.97 | 99.47 | 9.26 | 98.17 | 20.05 | 95.80 | 24.63 | 95.13 | 14.23 | 96.98 |
| $\mathcal{L}_{\mathrm{OMB}}$ | 0.43 | 99.87 | 6.48 | 98.55 | 15.29 | 96.38 | 28.19 | 94.02 | **12.46** | **97.29** |

**Comparision of different inference strategies.** Please refer to Table 11.

**Comparing $\mathcal{L}_{\mathrm{CE}}$ and $\mathcal{L}_{\mathrm{OMB}}$ across multiple datasets.** Please refer to Table 12.

**OOD prompt learning strategies.** Please refer to Table 13.

# D  ADDITIONAL RESULTS

**Different Backbone Architectures.** Please refer to Table 14. As shown in the results, TTOD still achieved the best performance under different backbones.

**Ordering of Testing Data.** Test-time adaptation methods are inevitably affected by the order in which the test samples arrive. To rigorously test this aspect, we randomly shuffled the order of the test data using three distinct seeds. We observed that our method exhibits robustness to changes in the ordering of test data. Specifically, across three experiments conducted on the ImageNet dataset, the AUROC scores were 97.34%, 97.2%, and 97.34%, respectively, demonstrating fluctuations of less than 0.2%. We report the average results from three random runs in our paper.

**Effectiveness on OpenOOD benchmark.** We use four popular ID datasets CIFAR-10/100 (Krizhevsky et al., 2009), ImageNet-200/1K (Deng et al., 2009). Following the OpenOOD benchmark (Yang et al., 2022), the OOD testing datasets are categorized into two groups: Near OOD and Far OOD. Specifically, for CIFAR-10/100 benchmarks, the Far OOD group includes MNIST (LeCun et al., 1998), SVHN (Netzer et al., 2011), Texture (Cimpoi et al., 2014), Places365 (Zhou et al., 2018), and the Near OOD group comprises CIFAR-100/10 and Tiny ImageNet-200 (Deng et al., 2009). For ImageNet-200/1K, the Near OOD group includes SSB-hard (Vaze et al., 2022), NINCO (Bitterwolf et al., 2023), and the Far OOD group comprises iNaturalist (Horn et al., 2018), Texture (Cimpoi et al., 2014), and OpenImage-O (Wang et al., 2022).

As shown in Table 15 and Table 16 on the OpenOOD benchmark (Yang et al., 2022), under the far-out-of-distribution (far-OOD) setting, our method consistently achieves the best results. Under the near-OOD setting, when using CIFAR-10, CIFAR-100, and ImageNet-200 as in-distribution (ID) datasets, our method performs comparably to AdaNeg, which utilizes an external dataset. This

Table 13: OOD prompt learning strategies across multiple datasets.

| Prefix | Classname | Update | iNaturalist | | SUN | | Places | | Texture | | Average | |
|---|---|---|---|---|---|---|---|---|---|---|---|---|
| | | | F↓ | A↑ | F↓ | A↑ | F↓ | A↑ | F↓ | A↑ | F↓ | A↑ |
| Random | Random | ✓ | 0.30 | 99.90 | 7.19 | 98.43 | 16.47 | 95.76 | 30.55 | 92.91 | 13.63 | 96.75 |
| Random | ID | ✓ | 0.30 | 99.90 | 7.82 | 98.28 | 18.84 | 95.35 | 34.31 | 92.28 | 15.32 | 96.45 |
| Manual | ID | x | 54.84 | 89.38 | 53.59 | 89.08 | 57.78 | 86.04 | 59.77 | 84.75 | 56.49 | 87.32 |
| Manual | ID | ✓ | 0.43 | 99.87 | 6.48 | 98.55 | 15.29 | 96.38 | 28.19 | 94.02 | **12.46** | **97.29** |

Table 14: Performance comparison on the ImageNet-1K benchmark with ResNet50 backbone.

| Method | iNaturalist | | SUN | | Places | | Texture | | Average | |
|---|---|---|---|---|---|---|---|---|---|---|
| | FPR95↓ | AUROC↑ | FPR95↓ | AUROC↑ | FPR95↓ | AUROC↑ | FPR95↓ | AUROC↑ | FPR95↓ | AUROC↑ |
| Neglabel | 2.6 | 99.29 | 22.62 | 95.05 | 47.71 | 90.0 | 42.85 | 89.80 | 28.95 | 93.53 |
| FA | 68.97 | 82.02 | 55.84 | 86.93 | 62.58 | 82.68 | 34.96 | 91.74 | 55.59 | 85.84 |
| AdaNeg | 1.07 | 99.63 | 13.22 | 96.78 | 35.11 | 93.44 | 25.61 | 94.67 | 18.75 | 96.13 |
| OODD | 3.72 | 99.05 | 30.80 | 93.66 | 60.06 | 83.76 | 48.58 | 88.55 | 35.79 | 91.25 |
| AdaND | 4.21 | 98.9 | 19.5 | 95.62 | 21.61 | 94.2 | **17.9** | **95.06** | 15.50 | 95.94 |
| TTOD(ours) | **0.73** | **99.75** | **9.39** | **98.16** | **18.49** | **95.25** | 26.88 | 94.21 | **13.87** | **96.84** |

demonstrates the effectiveness of our approach across diverse experimental settings. When our method employs the baseline detector MCM, which performs poorly at distinguishing ID and OOD samples, our approach exhibits performance degradation. This occurs in the most challenging near OOD setting using ImageNet-1k as the ID dataset. It is worth mentioning that this phenomenon also appears in the training-based method IDlike. Under such extreme difficulty, all methods face significant limitations. We attempted to replace the baseline detector with a more effective one, FA. The results show that when our method is combined with FA across all test cases, it achieves remarkable performance and consistently yields the best results. This demonstrates that our method remains effective even under extremely challenging conditions.

Table 15: Performance Comparison on OOD Detection Across Datasets. Lower FPR95 and higher AUROC are better. Best results are in **bold**, and the second-best results are underlined.

| ID Dataset | Method | External Data | Near OOD | | Far OOD | |
|---|---|---|---|---|---|---|
| | | | FPR95 ↓ | AUROC ↑ | FPR95 ↓ | AUROC ↑ |
| CIFAR 10 | MCM | × | 35 | 91 | 12.57 | 96.77 |
| | Neglabel | √ | 35.32 | 92.96 | 15.74 | 96.29 |
| | AdaNeg | √ | 32.38 | **94.01** | 7.31 | 98.28 |
| | OODD | × | 48.61 | 89.19 | 14.11 | 96.7 |
| | AdaND | × | 35.13 | 90.81 | 0.78 | 99.62 |
| | ours | × | **30.25** | 93.6 | **1.07** | **99.75** |
| CIFAR 100 | MCM | × | 91.01 | 70.53 | 73.27 | 79.66 |
| | Neglabel | √ | 77.54 | 71.9 | 59.66 | 79.84 |
| | AdaNeg | √ | 71.62 | 77.56 | 40.81 | 88.41 |
| | OODD | × | 91.52 | 70.23 | 69.48 | 79.88 |
| | AdaND | × | 77.35 | 70.5 | 22.45 | 92.53 |
| | ours | × | **52.34** | **82.33** | **0.21** | **99.64** |
| ImageNet-200 | MCM | × | 63.66 | 83.66 | 17.97 | 96.13 |
| | Neglabel | √ | 49.83 | 87.61 | 9.36 | 97.87 |
| | AdaNeg | √ | 41.48 | **88.76** | 9.79 | 98.05 |
| | OODD | × | 48.39 | 83.82 | 14.82 | 96.13 |
| | AdaND | × | 50.35 | 85.62 | 15.15 | 95.86 |
| | ours | × | 48.16 | 87.75 | **7.89** | **98.5** |

Table 16: Performance Comparison on ImageNet1k OOD Detection. Lower FPR95 and higher AUROC are better. Best results are in **bold**, and the second-best results are underlined.

| ID Dataset | Method | External Data | Near OOD | | Far OOD | |
|---|---|---|---|---|---|---|
| | | | FPR95 ↓ | AUROC ↑ | FPR95 ↓ | AUROC ↑ |
| ImageNet1k | MCM | × | 84.17 | 69.22 | 44.39 | 90.61 |
| | Neglabel | √ | 69.27 | 75.38 | 23.23 | 94.94 |
| | LoCoOp | × | 82.51 | 68.03 | 33.42 | 92.12 |
| | IDLike | × | 86.23 | 59.51 | 36.11 | 92.16 |
| | LocalPrompt | × | 77.91 | 73.44 | 28.6 | 93.71 |
| | FA | × | 69.21 | 77.97 | 25.78 | 93.67 |
| | AdaNeg | √ | 67.35 | 76.01 | 20.9 | 95.44 |
| | AdaNeg + FA | √ | 76.15 | 73.79 | 95.38 | 19.75 |
| | OODD | × | 73.83 | 67.08 | 24.81 | 91.87 |
| | OODD + FA | × | 60.06 | 77.31 | 23.52 | 92.21 |
| | AdaND | × | 70.93 | 73.93 | 20.45 | 94.07 |
| | AdaND + FA | × | 69.15 | 75.82 | 18.69 | 94.66 |
| | ours | × | 89.44 | 56.3 | 19.88 | 95.59 |
| | ours + FA | × | **56.53** | **83.33** | **13.62** | **97.05** |

