# OpenReview forum: "Beyond Visual cues: Harnessing Text signal for Test-Time OOD Detection"
_ICLR.cc/2026/Conference — ICLR 2026 Conference Withdrawn Submission_

### Official Review · Reviewer_8S5c · 2025-10-26

**Soundness:** 3
**Presentation:** 3
**Contribution:** 3
**Rating:** 6
**Confidence:** 3

**Summary:**

This paper targets open-world deployment and studies test-time (post hoc) OOD detection based on pre-trained vision–language models (e.g., CLIP). The authors note that most existing methods primarily cache/process visual features, leaving the adaptive potential of the textual modality underexplored. To address this, they propose Test-time Textual OOD Discovery (TTOD): without any external labels, TTOD leverages pseudo-labels from a base detector to optimize learnable OOD prompts online to mine OOD semantics, uses the clustering property of OOD to filter ID boundary samples misclassified as OOD (OOD Knowledge Purification, with the associated loss LOKP), and maintains high-quality OOD textual embeddings in an OOD Textual Knowledge Bank (OKB) for score calibration. Experiments on ImageNet-1K, CIFAR-100, and nine OOD datasets demonstrate the value of textual semantic mining for robust test-time OOD detection.

**Strengths:**

1. Table 9 shows that TTOD is not tied to a specific base detector and delivers significant gains when combined with multiple detectors such as GL-MCM, FA, NegLabel, and LoCoOp.

2. TTOD dynamically mines OOD textual semantics from the test stream: through continual optimization of learnable OOD prompts, the textual modality is directly adapted to the real OOD distribution without any external annotations.

3. &#x20;Initializing with the same template as the ID prompt and updating online at test time yields the most discriminative OOD textual embeddings and the best detection performance.

**Weaknesses:**

1. The paper introduces a coefficient  $\beta$  to balance the base score and the text-side calibration. Although the valid range and sensitivity curves for $\beta$ are provided, re-estimation is still required when deploying across datasets.

2. The algorithm requires batch-level OOD prompt optimization and OKB dynamic updates with test-time calibration, but it does not provide end-to-end comparisons with strong baselines in terms of throughput/latency/memory.

3. Although sensitivity analysis over $K \in [2^8, 2^{14}] $indicates robustness, the resource–performance trade-off for the default (e.g., 2048) and the overhead differences across retention strategies (RAND/FIFO/SA/DBR) are not finely reported.

**Questions:**

1. TTOD’s OOD prompts are initialized from an ID prompt template. While this design can quickly align the prompts with the data distribution, when the semantic gap between ID and OOD is very large, will the initial ID semantic bias slow convergence for OOD prompt optimization or even cause local optima?

2. The OKB capacity is fixed at K = 2048, but the rationale for choosing 2048 is not explained. When the test scale is very small, does 2048 lead to redundant storage (most embeddings having low discriminative semantics)? When the test scale is very large, does 2048 fail to cover all highly discriminative OOD semantics?

3. TTOD’s performance relies on sufficiently many highly discriminative OOD embeddings in the OKB, but scenarios with an extremely low OOD ratio are not validated. If each batch yields only limited OOD prompt semantics, the OKB may accumulate useful OOD knowledge slowly—does this cause a sharp drop in OOD detection accuracy in the early testing stage?

4. Table 14 reports experiments across different backbones; it is recommended to include more model scales, such as CLIP-L, CLIP-H, and RN101.

---

### Official Review · Reviewer_Nuyt · 2025-10-27

**Soundness:** 3
**Presentation:** 4
**Contribution:** 3
**Rating:** 6
**Confidence:** 4

**Summary:**

### Background
- This study focuses on test-time adaptation OOD detection methods.
- The authors state that most existing test-time OOD detection methods rely on storing representative visual features, while the adaptive potential of the text modality has been largely overlooked.

### Method
- They propose the Test-time Textual OOD Discovery (TTOD) framework.
    - At test time, TTOD optimizes the OOD prompts and keeps updating an OOD textual knowledge bank.
    - Besides, the authors develop a purification strategy to alleviate the impact of contaminated signals.

### Results
- TTOD consistently outperforms prior works and achieves SoTA performance.

**Strengths:**

- The paper is well-written with clear figures and tables.
- The method is clear and easy to understand.
- The experiments are comprehensive and TTOD consistently outperforms prior works.

**Weaknesses:**

### Major
- TTOD v.s. AdaNeg
    - TTOD shares a similar framework with AdaNeg (both have OOD textual features and a memory bank), but TTOD significantly outperforms AdaNeg. Could the authors explain what is the key advantage of TTOD over AdaNeg, and why using textual features is better than visual features?
    - Is it possible to combine TTOD and AdaNeg and achieve better performance?

- About implementation details.
    - Now that TTOD does not require external OOD labels, I think the framework relies on the OOD prompt initialization. In lines 287-289, the authors briefly explain how they do OOD initialization, but I'm kind of confused. Could the authors introduce the initialization in detail? Besides, how does TTOD perform with different initialization strategies?
    - How do the authors decide the four hyperparameters in TTOD?
    - How does TTOD perform with different batch sizes during test time?

### Minor
- (Line 481) A period is missing.
- (Line 454) Wrong quotation mask.

**Questions:**

Please see the Weaknesses section.

Besides, I’m really impressed that the average AUROC for OOD detection has already reached 97.29. What do the authors think can still be improved for OOD detection with CLIP?

---

### Official Review · Reviewer_pvpA · 2025-10-29

**Soundness:** 2
**Presentation:** 2
**Contribution:** 2
**Rating:** 4
**Confidence:** 4

**Summary:**

The paper introduces Test-time Textual OOD Discovery (TTOD), a framework that adapts CLIP at test time by learning OOD textual prompts directly from the data stream. It uses a base detector (MCM) for pseudo-labeling, optimizes prompts via a minority-balanced loss and a purification loss that exploits semantic clustering in text space, and maintains a fixed-size OOD Textual Knowledge Bank to calibrate scores. Evaluated on ImageNet-1K and CIFAR-100 with 9 OOD datasets, TTOD achieves 11.63% lower FPR95 and 4.29% higher AUROC than prior test-time methods.

**Strengths:**

- Unlike AdaND and OODD, which store and process visual features, TTOD operates exclusively on learnable text prompts. This leverages CLIP’s compact, concept-level text embeddings, enabling semantic-level adaptation without storing high-dimensional image features.

- TTOD achieves SOTA performance on both ImageNet-1K and CIFAR-100, reducing FPR95 by 11.63% and improving AUROC by 4.29% on average.

- TTOD operates in a fully unsupervised manner, requiring no OOD labels, external datasets, or predefined negative prompts whatsoever.

**Weaknesses:**

- The OOD Knowledge Purification method relies on the unrealistic assumption that “most OOD samples in a batch belong to the same high-level textual semantic cluster”. This premise fails by design in truly open-world settings, where OOD is inherently unpredictable.

- Pseudo-labels are derived from MCM + adaptive threshold λ using a fixed queue. Early misclassifications can poison the OOD prompts, causing irreversible drift toward ID semantics over time. No drift detection, confidence gating, or reset mechanism is proposed.

- The semantic clustering phenomenon and effectiveness of $L_{OKP}$ are empirically observed, not theoretically analyzed. No discussion of convergence properties, prompt initialization sensitivity, or conditions under which OOD clustering emerges.

- TTOD requires continuous optimization of learnable OOD text prompts during inference, introducing computational overhead and latency that may hinder real-time deployment in resource-constrained or low-latency applications.

- No experiments are conducted under near-OOD conditions, where OOD samples lie close to the ID decision boundary in feature space — a setting explicitly acknowledged as difficult in prior work and inherently more challenging than far-OOD.

**Questions:**

How sensitive is TTOD’s performance to (1) batch size and (2) the temporal ordering of test samples? Specifically, does the method remain stable under small batch sizes (e.g., B=1) or highly interleaved ID/OOD streams, and are such robustness analyses provided in the paper?

---

### Official Review · Reviewer_zxLK · 2025-10-31

**Soundness:** 3
**Presentation:** 2
**Contribution:** 2
**Rating:** 4
**Confidence:** 4

**Summary:**

This paper introduces Test-time Textual OOD Discovery (TTOD), a new framework for Out-of-Distribution (OOD) detection that operates during the test phase. The core idea is to move beyond adapting visual features and instead harness the adaptive potential of the textual modality in Vision-Language Models (VLMs). The method progressively builds a textual knowledge bank of OOD concepts by continuously optimizing a set of learnable OOD prompts. This optimization is guided by pseudo-labels generated from a base OOD detector. To improve the quality of these pseudo-labels, the framework includes a purification strategy that uses semantic clustering to separate misclassified in-distribution (ID) samples from true OOD samples. The discovered OOD textual embeddings are then stored and used to calibrate the predictions of the base detector. The authors conduct extensive experiments on ImageNet-1k and CIFAR-100 benchmarks, demonstrating that TTOD achieves state-of-the-art performance without requiring external labeled data.

**Strengths:**

1.The paper addresses the critical problem of test-time OOD detection. The core proposal to adapt the textual modality, rather than the more commonly targeted visual modality, is novel and well-motivated. It opens up an interesting new direction for research in VLM-based OOD detection.

2. The experimental results are comprehensive and impressive. The proposed method consistently outperforms a wide range of existing methods, including training-free, training-based, and other test-time adaptation techniques, across multiple standard benchmarks. The performance gains, especially in terms of FPR95, are significant.

3.The authors have conducted a thorough evaluation, including extensive ablation studies that validate the contributions of each component of their framework (L_OMB, L_OKP, and the OKB). The sensitivity analysis and comparisons with various base detectors further strengthen the paper's claims.

**Weaknesses:**

1. My main concern revolves around the framework's reliance on the base detector, S_base. Since the entire learning process is driven by pseudo-labels from S_base, the performance of TTOD seems intrinsically linked to the strength of this base detector. The paper shows impressive results when combined with already strong, state-of-the-art methods like FA. This naturally leads me to wonder how much of the performance lift comes from TTOD itself versus the powerful base it's built upon.

2. I also think about the design choice to freeze the ID prompts while only adapting the OOD prompts (as shown in Figure 1). I can see the motivation for focusing on discovering OOD knowledge, but it wasn't immediately clear to me why the ID prompts are kept static. In a real-world setting, one might expect some drift in the in-distribution data as well. Perhaps the authors could elaborate on the rationale behind this decision?

3. On a note of presentation, I found the terminology used to describe the method a bit confusing at times. The paper seems to switch between "TTOD," "Ours," and "TTOD + [Base Detector]". Since the base detector is such an integral part of the pipeline, it might make the paper easier to follow if a single, consistent naming convention like "TTOD(S_base)" were used throughout. This would help to constantly remind the reader of the method's structure and avoid any potential misunderstanding about it being a fully standalone detector.

4. Regarding Figure 2, it does not convey the feeling of an integrated or end-to-end architecture. On the contrary, it appears to depict a rather complex and multi-stage pipeline where components are bolted together: a base detector's output is used to create pseudo-labels, which then supervise a separate prompt optimization loop, whose results are then curated and used for a final calibration step. This visual representation seems to conflict with the paper's narrative of presenting a novel, cohesive framework. The fact that the diagram itself looks so fragmented highlights a potential lack of architectural elegance in the method's design, suggesting a collection of loosely coupled steps rather than a truly unified solution.

5. Building on my earlier point about the frozen ID prompts, I am concerned about how the system would handle shifts in the in-distribution data. The current setup seems to assume the ID distribution is stable. It would be a much stronger and more realistic evaluation to see how TTOD performs in a scenario with ID distribution shift, for example, by testing on a dataset like ImageNet-v2. If the base detector starts to fail on these shifted ID samples, it could generate noisy pseudo-labels that might severely degrade the OOD prompt learning.

6. Finally, I was curious about the practical implications of the proposed framework. The test-time adaptation process, with its optimization loop and knowledge bank updates, seems like it would introduce some computational overhead compared to a static detector. It would be very valuable to include some analysis of this, perhaps a discussion on the added latency or computational cost per batch.

**Questions:**

see Cons

---

### Note · Authors · 2025-11-12

**Comment:**

We sincerely thank all reviewers for their valuable time and constructive feedback.
After carefully considering the comments, we realized that several key clarifications and improvements are required to properly convey our ideas and results.
While most of the raised concerns can be addressed, we believe the current submission does not yet reflect the quality and clarity that the work deserves.
Therefore, we have decided to withdraw the paper to substantially revise it before resubmission.
We are grateful for the thoughtful reviews, which have greatly helped us identify critical directions for improvement.

**Withdrawal Confirmation:**

I have read and agree with the venue's withdrawal policy on behalf of myself and my co-authors.